# Osteopontin as a Biomarker in Chronic Kidney Disease

**DOI:** 10.3390/biomedicines11051356

**Published:** 2023-05-04

**Authors:** Satyesh K. Sinha, Michael Mellody, Maria Beatriz Carpio, Robert Damoiseaux, Susanne B. Nicholas

**Affiliations:** 1Department of Medicine, David Geffen School of Medicine, University of California, Los Angeles, CA 90095, USA; mariablcarpio@gmail.com; 2Division of Endocrinology, Molecular Medicine and Metabolism, Charles R. Drew University of Science and Medicine, Los Angeles, CA 90059, USA; 3Department of Bioengineering, Henry Samueli School of Engineering, University of California, Los Angeles, CA 90095, USA; mmellody@g.ucla.edu; 4Department of Molecular and Medical Pharmacology, David Geffen School of Medicine, University of California, Los Angeles, CA 90095, USA; rdamoiseaux@mednet.ucla.edu

**Keywords:** osteopontin, N-terminal osteopontin, chronic kidney disease, diabetic kidney disease, biomarker

## Abstract

Osteopontin (OPN) is a ubiquitously expressed protein with a wide range of physiological functions, including roles in bone mineralization, immune regulation, and wound healing. OPN has been implicated in the pathogenesis of several forms of chronic kidney disease (CKD) where it promotes inflammation and fibrosis and regulates calcium and phosphate metabolism. OPN expression is increased in the kidneys, blood, and urine of patients with CKD, particularly in those with diabetic kidney disease and glomerulonephritis. The full-length OPN protein is cleaved by various proteases, including thrombin, matrix metalloproteinase (MMP)-3, MMP-7, cathepsin-D, and plasmin, producing N-terminal OPN (ntOPN), which may have more detrimental effects in CKD. Studies suggest that OPN may serve as a biomarker in CKD, and while more research is needed to fully evaluate and validate OPN and ntOPN as CKD biomarkers, the available evidence suggests that they are promising candidates for further investigation. Targeting OPN may be a potential treatment strategy. Several studies show that inhibition of OPN expression or activity can attenuate kidney injury and improve kidney function. In addition to its effects on kidney function, OPN has been linked to cardiovascular disease, which is a major cause of morbidity and mortality in patients with CKD.

## 1. Introduction

Osteopontin (OPN) is a secreted, pleiotropic, multi-phosphorylated glycoprotein, first recognized as secreted phosphoprotein 1 (SPP1) in 1979 [1]. It was initially found in bone, but later research revealed that it is expressed in several tissues [2,3]. OPN mediates a multitude of biological processes and plays significant roles in biomineralization, and a number of physiological processes involved in cellular homeostasis, as well as in pathologies such as chronic inflammation and tumor biology [4,5,6]. OPN was also known as early T-lymphocyte activation 1 protein, and bone sialoprotein 1 [7] and was reported to activate immune cells, including T-cells, B-cells, macrophages, natural killer and Kupffer cells [1,8,9,10]. Currently, OPN is classified as one of the members of the small integrin-binding ligand N-linked glycoprotein family [11] and modulates cell signaling and connections with the matrix through interacting with integrins and CD44 receptors [7].

Among all tissues, kidneys have the greatest OPN content [12]. In normal kidneys, OPN is mainly expressed in the loop of Henle and the distal nephron [13], but following kidney damage, its expression is upregulated in all tubular segments and in the glomeruli by as much as 18-fold [13]. As such, it is not surprising that OPN has been shown to be involved in several diseases of the kidney [13]. In our own work, we showed that OPN has a critical role in the development and progression of diabetic nephropathy (DN) in experimental animal models of type 1 and type 2 diabetes and that N-terminal OPN (ntOPN) is implicated as the key mediator of these processes (work in progress). In this review, we describe the genetic structure and molecular signaling of OPN and discuss the roles of both OPN and thrombin-cleaved ntOPN as potential biomarkers of several forms of chronic kidney disease (CKD), especially DN, which presents clinically as diabetic kidney disease (DKD), and is the most common cause of CKD and kidney failure [14,15].

### 1.1. Molecular Structure and Function

The chromosome location of the OPN gene varies by species; for example, the gene in human, and mice is located on chromosomes 4, and 5, respectively [16]. A genome-wide association of a European CKD population revealed two replicated loci, one upstream of *SPP1,* and another mapping onto *KLKB1* encoding pre-kallikrein, which is involved in blood pressure control, inflammation, cancer, and cardiovascular disease [12]. An important paralog of *KLKB1* is *F11*, which is associated with OPN expression in several tissues. While the study could not answer whether these variants were responsible for the observed association in their study, the authors speculated that new bioactive OPN fragments could be generated by activated kallikrein. Further, single nucleotide polymorphisms (SNPs) of OPN have been associated with several autoimmune conditions. SNPs in the OPN gene have been associated with systemic lupus erythematosus (SLE), multiple sclerosis, inflammatory bowel diseases, and type 1 diabetes [17]. Both the NCBI and UniProt databases [9] show that the OPN primary transcript undergoes alternative splicing, resulting in at least five spliced variants: OPNa (which contains all coding exons), OPNb (which lacks exon 5), OPNc (which lacks exon 4), OPN4 (which lacks both exon 4 and exon 5), and OPN5 (which adds an exon by incorporating a region from intron 3) [9,18]. All OPN-spliced variants incorporate several highly conserved domains including an arginine-glycine-aspartic acid (RGD) recognition sequence (glycine-arginine-glycine-aspartic acid-serine (GRGDS), a SVVYGLR (in human and SLAYGLR in murine OPN) sequence, which is a thrombin cleavage site [9,19] (Figure 1A). In addition to spliced variants of OPN mRNA and proteolytic cleavage of the translated OPN protein, OPN undergoes several post-translational modifications, such as serine/threonine phosphorylation, glycosylation, and transglutamination [20]. Although the precise impact of these modifications in CKD is not well understood, they nevertheless serve as a vital mechanism for regulating and altering the function of OPN in various other pathologies (Table 1).

The central section of OPN contains sequences that can interact with several different integrins [36], as well as CD44, which forms an adhesion complex that promotes cell migration [37]. All animal lineages have high levels of integrins, which function as a class of noncovalently connected heterodimeric transmembrane adhesion receptors through various combinations of α and β subunits [36,38,39], and OPN–integrin interactions are both extremely common and important. OPN participates in many integrin-related signal transduction events (Figure 1B). For example, integrin αvβ3 binding of OPN activates the signaling pathways of focal adhesion kinase (FAK), extracellular signal-regulated kinase (ERK)1/2, and nuclear factor (NF)-κB leading to cellular migration [39,40,41]. This interaction also regulates cellular proliferation and survival via signal transducer and activator of transcription-3 signaling pathways [40,42,43]. Through the activation of the phosphatidylinositol (PI)3 kinase/AKT/mammalian target of rapamycin signaling pathway, an OPN–integrin–αvβ3 interaction also enhances proliferation and lowers the apoptosis pathway [44,45]. By facilitating cell migration via association with integrins α9β1 and α4β1, the cryptic RGD-integrin binding site within ntOPN has served as a chemoattractant for stem and progenitor cells. ntOPN also binds to α4β1 and α9β1 integrins and exerts a strong pro-inflammatory effect [46]. Unlike other integrin receptors for OPN, α9β1 recognizes only ntOPN and not full-length OPN [47]. Kale et al. identified that the interaction of OPN with the α9β1 integrin activates ERK and p38 signaling to stimulate cyclooxigenase-2 expression in macrophages, promoting tumor angiogenesis in melanoma [48]. Further, it has been established that the αvβ3–OPN complex activates FAK and AKT, boosting pulmonary artery smooth muscle cell proliferation, and improving vascular remodeling [49,50]. In epidermal growth factor receptor mutant non-small cell lung cancer, the interaction of integrin αvβ3 with OPN promotes acquired resistance to tyrosine kinase inhibitors of the growth factor receptor by activating the downstream FAK/protein kinase B and ERK signaling pathways [51].

### 1.2. N-Terminal Osteopontin (ntOPN)

The full-length OPN (flOPN) protein undergoes considerable post-translational modification such as phosphorylation and glycosylation. It is also cleaved by various proteases, including thrombin, matrix metalloproteinase (MMP)-3, MMP-7, cathepsin-D and plasmin to produce several ntOPN species [32,52]. ntOPN derived from cleavage of OPNa isoform has been studied more extensively than ntOPN derived from OPNb and OPNc, which appears to mediate more pathobiological effects rather than homeostasis which may be a consequence of its enhanced affinity with integrin receptors [20,33,53] as its RGD motif is sterically more accessible. For example, it has been shown by Boggio et al. [54] that OPNa-derived ntOPN plays a significant role in multiple sclerosis by boosting the production of interleukin (IL)-17 and IL-6. In a rat model of ischemic cardiomyopathy, ntOPN also hastens heart damage [55,56], and research in individuals with hypertension found a substantial correlation between ntOPN and carotid atherosclerotic plaque inflammation and plaque instability [31]. ntOPN has also been linked to alcoholic hepatitis in mice [57]. In addition, a recent study using a murine model of ischemic cardiomyopathy revealed that ntOPN increased the synthesis of type III collagen in cardiac fibrosis [58].

The pathologic role of ntOPN has been described in several forms of CKD. Gang et al. [59] reported that ntOPN in the urine of patients with immunoglobulin A nephropathy (IgAN) correlates with the level of albuminuria. In another study conducted in patients with lupus nephropathy (LN), the level of plasma ntOPN was significantly higher compared to non-LN patients. Moreover, urine ntOPN correlated positively with urine thrombin activity. The authors suggested that urine ntOPN is associated with renal inflammation and may be a good prognostic marker for LN groups [60]. Additionally, ntOPN was detected in patients with minimal change disease (MCD), but not in controls. As a result of the involvement of ntOPN in several kidney diseases, ntOPN has also been considered as a biomarker in these conditions, including DKD, which is the focus of our ongoing studies.

## 2. Potential of Osteopontin as a Biomarker in Chronic Kidney Disease

OPN was initially demonstrated to be an important potential biomarker mostly for the diagnosis of several cancers, such as non-small cell lung cancer, prostate, glioblastoma, malignant glioma [61,62,63]. However, a large number of studies have shown increased OPN mRNA and OPN protein levels in several forms of CKD including DKD, LN, IgAN, tubulointerstitial nephritis, glomerulonephritis, acute ischemic renal injury, interstitial inflammation, fibrosis, and hydronephrosis [13,64,65,66,67,68] (Table 2). These studies indicate a correlation between high levels of OPN expression, which can be quantified using commercially available sandwich enzyme-linked immunosorbent assays [69], with kidney function measured by proteinuria, reduced creatinine clearance, fibrosis, and macrophage and T-cell infiltration [13,64,65,70] (Figure 2). Subsequently, OPN was later reported to be a powerful predictor of CKD including incipient DN [66,71].

### 2.1. Diabetic Kidney Disease (DKD)

DKD is the most common cause of CKD and kidney failure [15,83,84]. In fact, DKD is a significant medical problem for individual patients and a major socioeconomic burden as a result of the high morbidity and mortality rates [85]. It is generally known that the approach to managing DKD entails regulating hyperglycemia, dyslipidemia, blood pressure, and the use of renin-angiotensin-aldosterone system (RAAS) inhibitors in the early stages of DKD and more recently, sodium glucose co-transporter-2 (SGLT-2) inhibitors, glucagon-like peptide (GLP)-1 receptor agonists and non-steroidal mineralocorticoid receptor antagonists (nsMRAs) [86,87]. Yet, the prevalence of DKD remains high despite intensive treatment with hyperglycemia control, blood pressure control, use of RAAS blockers [88] and the newer SGLT-2 inhibitors, GLP-1 receptor agonists and nsMRA therapies [89,90]. Since DKD remains asymptomatic for many years, risk verification and earlier diagnosis can potentially facilitate more timely, optimal care to prevent both the onset, and progression of DKD and for risk stratification that are crucial in limiting the personal and societal costs associated with DKD.

The most commonly used biomarkers for DKD are albuminuria and estimated glomerular filtration rate (eGFR), but these indicators do not optimally predict the progression of the disease and in many instances, may not be reliable in diagnosing this condition. For example, in the earliest microalbuminuria stage, multifactorial therapy may improve renal function [91] but this indicator has significant limitations [92]. Some patients with diabetes mellitus do not manifest excessive urinary albumin loss but still develop DKD and may progress to kidney failure. In fact, a study of 168 patients with either type 1 or type 2 diabetes mellitus revealed that 106 patients had histopathologic alterations similar to those associated with DKD, but up to 20% of these individuals had either no albuminuria or low eGFR throughout their lifetime. Additionally, structural changes were extremely varied and included nearly all DKD histopathologic groups [93]. The landmark United Kingdom Prospective Diabetes Study, multicenter, randomized clinical trial of 5102 patients with newly diagnosed type 2 diabetes mellitus also showed that whereas 14% of patients with DKD did not have excess albuminuria, 28% of patients developed moderate to severe renal impairment [94]. Due to the fact that some individuals may present with albuminuria while others may experience eGFR decline without signs of albuminuria, albuminuria and eGFR decline may not be linked and may represent different pathologic processes of DKD [94]. Other than creatinine-based eGFR, serum cystatin C-based-eGFR is an alternative measure of eGFR and is thought to be more accurate than serum creatinine-based eGFR in discriminating type 2 diabetes mellitus patients with reduced eGFR from those with normal eGFR [95,96]. Several other biomarkers have been developed to assess the presence of DKD. Urinary angiotensinogen and angiotensin converting enzyme (ACE)2 [97], plasma copeptin [98], plasma endostatin, serum amyloid [99], urinary neutrophil gelatinase-associated lipocalin, serum tumor necrosis factor receptor 1, and 2 [100] are a few of the biomarkers that have been examined [92]. However, currently, none of these novel biomarkers has been validated and approved by the United States Food and Drug Administration (FDA) for routine use in clinics mostly due to insufficient diagnostic sensitivity and specificity as demonstrated in clinical trials. Although our knowledge of DKD has significantly improved over the past several years, better diagnostic tools, and prognostic markers are still desperately required to provide early identification of DKD and OPN, and likely ntOPN, may be a potential contender in this line of investigation.

During the last decade, a number of studies analyzed the role of OPN in the pathogenesis of DKD and reported high expression of OPN in the tubular epithelium of the renal cortex and in glomeruli in rat and mouse models of DN [72,73]. We subsequently examined whether OPN might be a critical mediator of the glomerular pathologic lesions of DN. We treated type 2 diabetic db/db mice with either insulin or one of two thiazolidinediones (TZDs), rosiglitazone, and pioglitazone to achieve similar fasting plasma glucose levels [65]. The urine albumin-to-creatinine ratio and glomerular OPN expression were increased in diabetic mice but were significantly reduced by the TZDs more than by insulin. We then generated streptozotocin-induced experimental, and genetic models of type 1 (Ins2^Akita^) and 2 (Lepr^db/db^) diabetic mice on the background of OPN-null and OPN-wild-type mice. In each case, OPN deletion decreased albuminuria, glomerular mesangial area, fractional volume of expansion, and expression of glomerular collagen IV, fibronectin and transforming growth factor (TGF)-β in the diabetic mice compared with their respective controls. In cultured mouse mesangial cells, the TZDs but not insulin, decreased angiotensin II-induced OPN expression, while recombinant OPN upregulated TGF-β, ERK/mitogen-activated protein kinases (MAPK), and Jun N-terminal kinase (JNK)/MAPK signaling, which have been shown to be involved in the pathology of DN. These studies strongly suggested that OPN expression enhanced glomerular damage, likely through the expression of TGF-β, while the deletion of OPN protected against disease development and disease progression, suggesting that OPN might serve as a therapeutic target [65]. Apart from our research, a limited number of studies has explored the involvement of OPN in DKD. Yan et al. [101], for instance, found a correlation between plasma levels of OPN and the existence and gravity of nephropathy and coronary artery disease in individuals with type 2 diabetes mellitus. According to Lorenzen et al., OPN was shown to be involved in the development of albuminuria by modulating podocyte signaling and motility [102]. Yamaguchi’s research suggested that plasma OPN levels may increase with the advancement of DKD, indicating that the concentration of OPN in plasma could serve as a potential diagnostic predictor for diabetic kidney failure [103]. OPN overexpression and macrophage recruitment may also be involved in the tubulo-interstitial damage in DN because it was linked to significant macrophage buildup in the renal interstitium in DN [72,74]. Moreover, a positive correlation was observed between OPN and urinary N-acetyl-β-D-glucosaminidase/creatinine levels in patients with leptospirosis suffering from acute kidney injury [104]. Further, the use of OPN and neutrophil gelatinase-associated lipocalin together has been demonstrated to improve the detection of drug-induced kidney injury in a murine model [105]. Taken together, these findings imply that OPN has the potential to be utilized as a marker for both kidney glomerular and tubular injury. In addition, podocyte signaling and motility may be altered in OPN null mice to protect them against diabetes-induced albuminuria and renal impairment [102].

In humans, the presence and severity of DKD are independently correlated with plasma OPN levels [101]. Of particular interest is the connection between the renin angiotensin system (RAS) and OPN in DKD. There is compelling evidence in the literature in support of this observation: perindopril, a long-acting ACE inhibitor, dramatically reduced diabetes-induced OPN expression and macrophage accumulation in the kidney interstitium of diabetic rats [72]. Li et al. showed that upregulation of OPN expression may play a role in tubulointerstitial injury associated with DN, and RAS blockade by ramipril may confer renoprotection by decreasing OPN expression in non-insulin-dependent DN [75].

### 2.2. Lupus Nephritis

SLE is a multisystemic inflammatory rheumatic disease that often shows periods of flares followed by remission. Autoantibodies are produced as a result of immune system dysregulation; these antibodies combine with antigens to form complexes that are then deposited in different organs [106,107]. One of the most severe complications of SLE is LN [108]. Numerous investigations have shown that OPN levels are elevated in the plasma and urine of patients with SLE and that the increase may be related to the development and clinical symptoms of the disease [109]. Plasma OPN concentration was shown to be considerably greater in patients with SLE and renal impairment than in healthy controls. Additionally, elevated OPN showed a favorable correlation with both the IL-18 level and the SLE Disease Activity Index (SLEDAI) [76]. In another study, full-length OPN and ntOPN concentrations in urine and plasma were examined in LN patients; however, there was no difference in OPN concentration in urine between patients and healthy controls. However, patients with LN had considerably greater plasma full-length OPN and ntOPN levels [60]. Moreover, there was a favorable correlation between urine thrombin activity and urine ntOPN, but not with urine full-length OPN. Additionally, the study found no correlation between SLEDAI and eGFR and plasma and urine full-length OPN and ntOPN concentrations. According to the authors, urine ntOPN was related to renal inflammation and may be a reliable prognostic indicator for LN. Another investigation revealed that intrarenal macrophage infiltration and OPN expression were positively associated, with OPN expression in patients with LN, which was higher in these patients than in healthy controls [77]. Furthermore, OPN levels were higher in patients with active LN than in those with inactive LN [110].

Additionally, studies indicate that OPN-SNPs may be related to LN [111]. A genetic study showed that the T allele of the rs1126616 polymorphism was associated with renal dysfunction in SLE patients [111]. Similarly, the rs1126616 SNP was genotyped in a patient with SLE and showed that the frequency of CT and TT genotypes were higher in SLE patients with LN than in the control group [110]. This suggests that this polymorphic T allele is a risk factor for renal impairment in SLE patients. In another study, OPN gene 9250 polymorphism appeared to be associated with the susceptibility to LN in the southern Chinese Han population [112].

### 2.3. Immunoglobulin A Nephropathy (IgAN)

IgAN is a kidney disease caused by the accumulation of the immunoglobulin A antibody and is the most common form of primary glomerulonephritis, worldwide [113]. In a cohort of biopsy-proven glomerulopathy, IgAN, membranous nephropathy, and LN, OPN had an accuracy of 87% in distinguishing IgAN from other glomerulopathies, and thus appears to be a valuable biomarker [66]. Other reports suggest that OPN is indeed involved in the development of this type of nephropathy [59]. In patients with mild glomerulonephritis, the amount of OPN in their urine increases rapidly in response to a small amount of urinary protein [114]. This could be a sign of focal and segmental glomerulosclerosis (FSGS; a type of glomerular disease marked by glomerular fibrosis), since urinary OPN is often lower in IgAN [59]. Studies have also found that OPN and CD44 receptor are highly expressed in cells in areas of tubulointerstitial injury in IgAN [78]. Another study conducted in children with IgAN showed that the urine level of OPN was higher in patients compared to healthy controls and was associated with high OPN-to-creatinine ratio (OCR) [79]. Additionally, in these patients, the increased OPN mRNA expression was correlated with macrophage infiltration [79].

### 2.4. Autosomal Dominant Polycystic Kidney Disease (ADPKD)

Osteopontin has also been identified as a urinary biomarker for ADPKD progression. This pathology is characterized by the presence of fluid-filled renal cysts that may lead to end-stage renal disease. Kim et al. demonstrated that urinary OPN excretion levels were lower in rapid progressors than in slow progressors, suggesting that it may be a useful urinary biomarker for predicting ADPKD progression [80].

### 2.5. Minimal Change Disease (MCD)

MCD is a disorder that affects the kidney, causing selective proteinuria, hypoalbuminemia, hypercholesterolemia, and the absence of glomerular immune deposits or cellular infiltrates in kidney biopsies [115]. In MCD, podocyte injury is observed [116]. In some cases, the disease can progress to FSGS. The association between OPN and these pathologies was investigated in several studies. One study found that the urinary OCR in children with MCD and FSGS was higher than in the control group [79]. Additionally, the OCR was higher in FSGS patients than in MCD patients and was associated with interstitial changes and mesangial expansion. However, there is some disagreement about whether MCD patients have higher or lower levels of OPN in their urine. A positive correlation of OPN mRNA expression in proximal tubules and urinary OPN excretion was shown in MCD patients [59] in one study while another demonstrated that there was no significant difference in urine OPN levels between MCD patients and healthy controls [60]. Nevertheless, the plasma OPN concentration was higher in the MCD group. Similarly, Gang et al. [59] obtained opposite results and showed that urinary excretion of OPN in patients with MCD did not differ significantly from healthy controls. The group found that MCD patients have a higher rate of urine fragments that have a size of 34 kDa (ntOPN). This fragment is also associated with higher levels of albuminuria [59]. Another study, however, found no OPN expression in renal biopsy specimens from patients with MCD [81]. Therefore, controversies remain regarding the utility of OPN as a biomarker of MCD.

### 2.6. Membranous Glomerulonephritis

In adults, membranous glomerulonephritis (MGN) is the second most common cause of nephrotic syndrome, after FSGS. It is a slowly progressive disease of the kidney with a high incidence of proteinuria and edema (with or without kidney failure) [117,118]. Primarily, MGN is a disorder of autoimmune origin, but can also be a result of tumors, infectious or other autoimmune complications [117]. MGN is associated with the formation of immune complexes in subepithelial sites, which may lead to glomerular damage [119]. It is also possible that autoantibodies bind to podocyte membrane antigens and lead to subepithelial deposition of immune complexes [120]. Although the autoimmune basis of MGN is not fully understood, a wide spectrum of immune mediators is being investigated, including OPN. Recent studies demonstrated that patients with progressive and nonprogressive MGN had an overexpression of OPN in the proximal tubules [81,82]. Moreover, in MGN, a strong correlation between OPN mRNA and macrophage infiltration has been demonstrated. In a murine model, high expression of OPN in kidneys was associated with increased infiltration of macrophages, as well as CD4+ and CD8+ T cells. Overexpression of OPN in kidneys has also been correlated to activation of NF-ĸB, increasing the expression of proinflammatory cytokines, which can contribute to glomerular damage and elevated OPN expression [82]. Thus, it was suggested that OPN could be a potential predictor of primary MGN progression.

### 2.7. End Stage Renal Disease (ESRD)

ESRD or kidney failure has been linked to OPN, which has been shown to play a role not only in the progression of kidney disease but also in the development of vascular calcification associated with advanced CKD and in patients with kidney failure. A study found that serum OPN levels in peritoneal dialysis patients were positively correlated with aortic stiffness, as assessed by carotid-femoral pulse wave velocity [121]. In hemodialysis patients, the level of plasma OPN was associated with the presence of native arteriovenous fistulas stenosis requiring intervention, supporting its potential as a diagnostic biomarker [122]. In patients with kidney failure, OPN and tumor necrosis factor receptor type II (TNFR2) have a strong correlation with serum thrombomodulin (sTM), a well-established marker of endothelial injury. This association suggests a possible link between endothelial damage, inflammation, and vascular remodeling mediated by OPN and TNFR2. Moreover, the progression of radial artery calcifications in ESRD patients is thought to be related to a compensatory increase in sTM, which is indicative of endothelial injury. sTM plays a critical role in endothelial function and has potential clinical and prognostic application [123]. In a study of patients on maintenance hemodialysis, OPN levels were found to have significant positive correlations with intact parathyroid hormone, indicating that OPN may play an integral role in the mineral bone disorder axis. As such, the data suggest that it may be important to monitor OPN when managing patients on maintenance hemodialysis [124].

## 3. Therapeutic Approaches to Modulate OPN Expression and Function

Despite a significant body of research associating OPN and ntOPN with CKD, there are currently no FDA-approved OPN or ntOPN-targeting therapies available. However, several therapeutic approaches have been suggested that modulate OPN expression and function. In this section, we describe the current state-of-the-art of therapies, mainly in various cancers, involving OPN inhibition via antibodies and peptides, RNA interference (RNAi), and finally small molecule drugs. Knowledge of these approaches can potentially drive therapies targeting OPN and ntOPN in CKD (Figure 3).

### 3.1. Blocking of OPN and Its Receptors, Integrins and CD44, by Specific Antibodies/Peptides

Indirect inhibition of OPN: Liver X receptors (LXRs) have been identified as important lipid-dependent regulators of glucose metabolism and immune functions in leukocytes [125] and are thus of high importance. Synthetic LXR ligands can inhibit cytokine induced OPN expression in macrophages [126]. OPN expression was also reported to be suppressed by synthetic agonists for LXRs in a murine model of streptozotocin-induced diabetes. The LXR agonist significantly reduced macrophage infiltration, mesangial matrix buildup, and interstitial fibrosis without changing blood glucose levels or urinary albumin excretion. In the renal cortex, LXR activation reduced the gene expression of inflammatory mediators, including OPN. By blocking the activating protein-1-dependent transcriptional activation of the OPN promoter in proximal tubular epithelial cells in vitro, LXR activation reduced OPN expression. These findings imply that LXR agonists may have the ability to treat several forms of CKD, such as DKD, by inhibiting renal OPN [127]. For example, ntOPN function in mice has been demonstrated to be blocked by the polyclonal antibody, M5Ab, directed against a synthetic peptide (a cryptic epitope of OPN exposed by thrombin cleavage, VDVPNGRGDSLAYGLRS, M5 peptide) [33,128].

### 3.2. Employment of RNAi against OPN as a Potential Therapeutic Strategy

Inhibition of OPN via RNAi has been explored as a potential intervention for several therapeutic areas. OPN is overexpressed three-fold in kidneys of rats with ethylene glycol-induced hyperoxaluria and calcium oxalate nephrolithiasis relative to healthy rats [129]. It was found that in vivo transfections of OPN silencing RNA (siRNA) significantly reduced OPN mRNA levels and renal crystal deposition. In a 2022 study, exosomes electroporated with OPN siRNA were injected into mice with tetrachloride-induced liver fibrosis [130]. By reducing OPN mRNA levels in the liver, significant reductions in collagen deposition and α-Sma were observed, along with improved liver function. RNAi has also been used to silence OPN in several models of cancer. In primary and metastatic melanoma cell lines, it was found that siRNA suppression of OPN reduced cell proliferation and lowered expression of galectin-3, a marker of cancer stemness [4]. siRNA silencing of OPN in T24 bladder cancer cells decreased cell mobility and production of MMP-2 and MMP-9 migration-related proteins and also increased apoptosis rates. Apoptosis increase was correlated with increases in caspase-3, caspase-8, caspase-9, and p53 following OPN-RNAi treatment [131]. In a separate study of the Caki-1 renal carcinoma cell line, siRNA silencing of OPN resulted in induction of apoptotic peptidase activating factor 1, activation of caspase-3, decrease in b-cell lymphoma-2 (Bcl-2), and increase in Bcl-2-associated X protein [132], although the precise mechanism of this silencing of mitochondrial function is still under investigation. These results provide evidence of the role of OPN inhibition in triggering apoptosis in renal cancer cells that may serve as a paradigm for similar therapies in CKD.

### 3.3. Targeting OPN Using Small-Molecule Inhibitors

Given their relative ease in manufacturing and distribution, several small molecule drugs have been suggested as OPN inhibitors for a variety of disease conditions, but none has yet made it successfully into the clinic. In 2016, IPS-02001 (6,7-Dichloro-2,3,5,8-tetrahydroxy-1,4-naphthoquinone) was found to inhibit the interaction between OPN and integrin α_v_β_3_, reducing osteoclast formation in vitro [133]. Murine models of osteoporosis that were treated with IPS-02001 resulted in significant amelioration of bone loss compared to untreated mice. Given potential differences in the ligand efficiency of this small molecule and OPN, stark differences in overall effectiveness and potency can be expected. Moreover, the molecule is not an optimal drug chemotype and further optimization into a well-behaved and stable compound would be challenging. Other small molecules that act through the OPN–integrin axis are bisphosphonates [133]. In a prostate cancer model investigating α_v_β_3_–OPN blockade, bisphosphonates were found to inhibit the subsequent Rho GTPase activation, thus attenuating CD44/MMP-9 binding and reducing prostate cancer cell migration [134]. Downregulation of OPN through blockade of NR4A2 and Wnt signaling in colon cancer has been identified as an important component of cyclooxygenase-2 inhibitors such as parecoxib [135]. Andrographolide, a labdane diterpenoid, has been found to reduce expression of OPN and downregulate PI3 kinase/AKT signaling, resulting in breast cancer cell apoptosis and tumor reduction in mouse models [136]. Further work is needed before these compounds can be evaluated in clinical settings.

## 4. Conclusions

OPN, especially ntOPN, has been shown to play an important role in the development and progression of CKD. Both OPN and ntOPN have been implicated in renal inflammation and fibrosis, which are key pathological features of CKD. As such, OPN and ntOPN may serve as potential biomarkers of CKD, particularly DKD. Unfortunately, the level of albuminuria or eGFR in patients with DKD may not correlate with disease development or disease progression. There is evidence to support OPN, and especially ntOPN, in the underlying pathologic mechanisms of kidney diseases, such as DKD, and blockade of OPN may ameliorate DKD. Therefore, it is reasonable to assume that targeting OPN or ntOPN could be potential therapeutic strategies in some forms of CKD. In this regard, several approaches have been proposed, including antibodies, small molecule inhibitors, and gene therapy. However, these approaches are still in the preclinical stage and require further investigation to determine their safety and efficacy in humans.

## 5. Future Direction

Recent studies have suggested that OPN and ntOPN may serve as a biomarker for CKD, which is a frequent and commonly progressive condition that affects the kidneys. While preclinical studies have shown promising results in targeting OPN or ntOPN for the treatment of various cancers, further studies are needed to clarify the precise mechanisms by which OPN and ntOPN contribute to the development and progression of different forms of CKD. This could involve investigating the CKD signaling pathways and cellular processes, as well as specific interactions between OPN and other molecules within the kidney. In addition, more studies are needed to determine the safety and efficacy of targeting OPN and/or ntOPN in humans with CKD. This will require carefully designed clinical trials with appropriate patient populations and endpoints.

## Figures and Tables

**Figure 1 biomedicines-11-01356-f001:**
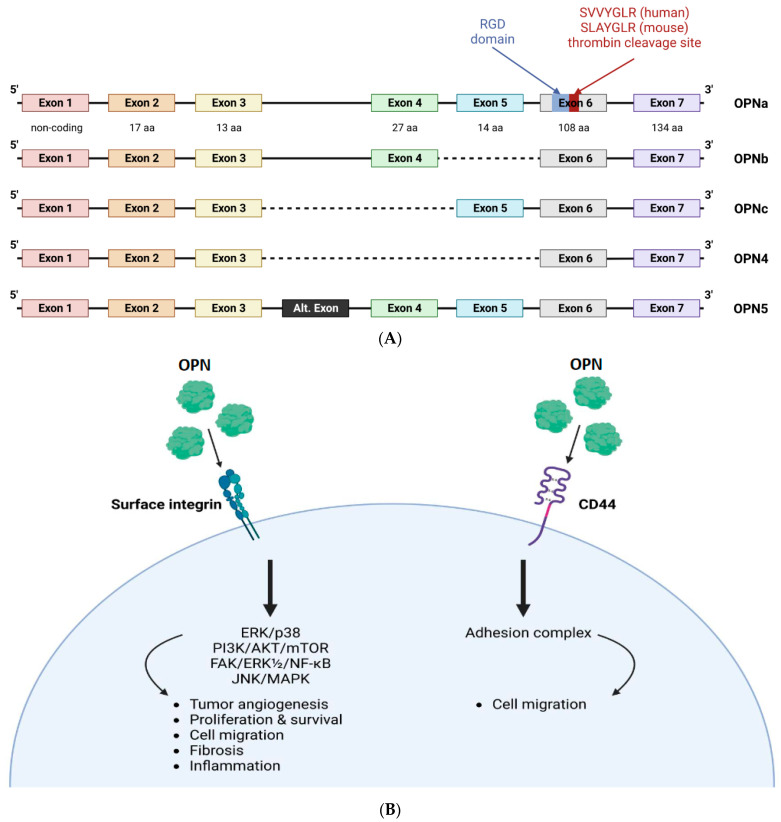
Structure of osteopontin (OPN) and OPN-mediated signaling pathways. (**A**). The figure shows the structure of OPN. OPN primary transcript undergoes alternative splicing, resulting in at least five spliced variants: OPNa (which contains all coding exons), OPNb (which lacks exon 5), OPNc (which lacks exon 4), OPN4 (which lacks both exon 4 and exon 5), and OPN5 (which adds an exon by incorporating a region from intron 3. All OPN-spliced variants incorporate several highly conserved domains, including an arginine-glycine-aspartic acid (RGD) recognition sequence (glycine-arginine-glycine-aspartic acid-serine (GRGDS), a SVVYGLR (in human and SLAYGLR in murine) sequence and a thrombin cleavage site. (**B**). The figure shows OPN-mediated signaling pathways. The pathological effects of OPN are mediated through the engagement of the receptors, integrins, and CD44. Ligation to these receptors results in important proinflammatory functions allowing OPN to mediate the activation of several pathways such as cell survival, cell proliferation, angiogenesis, migration, and fibrosis. Abbreviations: OPN- osteopontin; ERK/p38- extracellular signal-regulated kinase/p38; PI3K/AKT/mTOR- phosphatidylinositol-3-kinase/Ak strain transforming/mammalian target of rapamycin; FAK/ERK^1/2^/NF-κB- focal adhesion kinase/extracellular signal-regulated kinase 1 and 2/nuclear factor Kappa B; JNK/MAPK- jun N-terminal kinase/mitogen-activated protein kinase.

**Figure 2 biomedicines-11-01356-f002:**
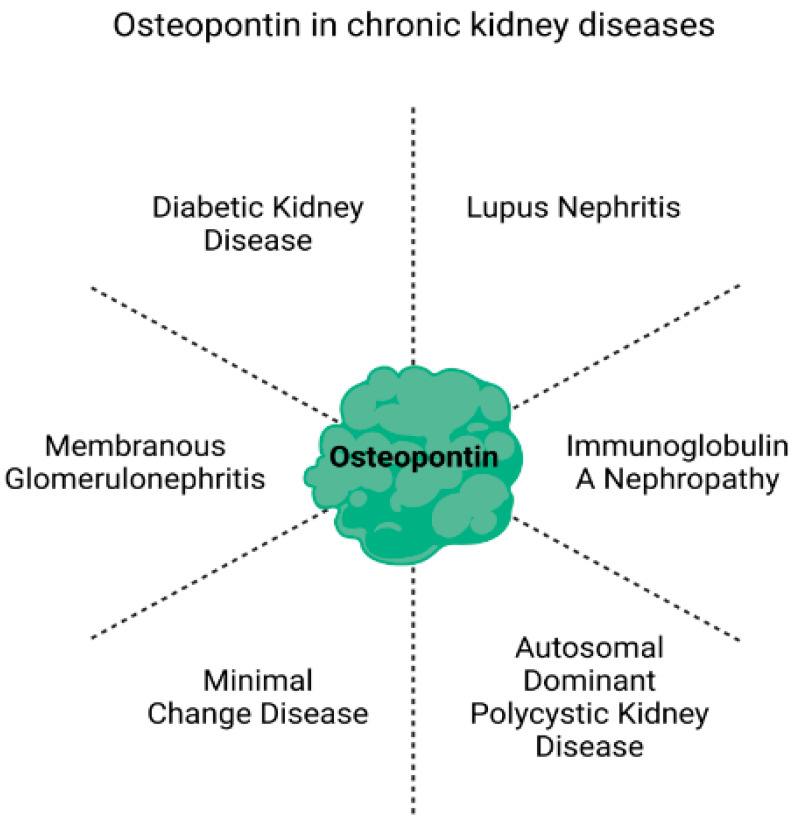
Osteopontin and chronic kidney disease. Increased levels of urine or plasma OPN have been associated with several forms of chronic kidney disease including diabetic nephropathy, which may present clinically as diabetic kidney disease, as well as lupus nephritis, immunoglobulin A nephropathy, minimal change disease, membranous glomerulonephritis, and autosomal dominant polycystic kidney disease.

**Figure 3 biomedicines-11-01356-f003:**
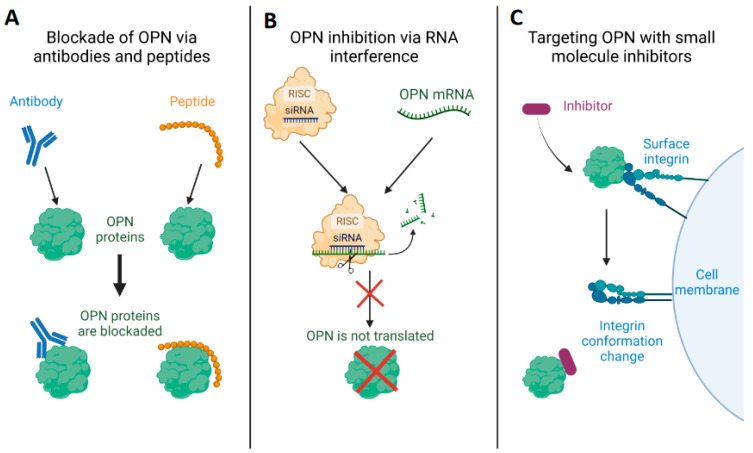
Summary of therapy modalities targeting OPN. (**A**). Translated OPN has been directly blockaded in numerous disease models using antibodies and small peptides. (**B**). In other work, OPN has been inhibited via RNAi approaches. (**C**). Small molecules have been explored to interrupt pathogenic interactions between OPN and surface integrins. OPN—osteopontin.

**Table 1 biomedicines-11-01356-t001:** Modifications of osteopontin and their functional characteristics.

Modifications of OPN		Functional Characteristics	References
Post-transcription modifications	OPNa	1. In a murine model of post-ischemic neovascularization, OPNa significantly boosted macrophage migration	[21]
		2. In non-small cell lung cancer (NSCLC), OPNa promoted cell migration, proliferation, colony formation, and invasion.	[22]
		3. OPNa was shown to significantly increase functional collateral vessel formation in vivo	[21]
	OPNb	1. In NSCLC, OPNb had a less significant effect as a modulator of proliferation, colony formation, and invasion.	[22]
		2. The overexpression of OPNb spliced variants in endometriotic cells activated the PI3K and NF-κB pathways. This led to endothelial mesenchymal transformation, cell migration, proliferation, morphological changes, and actin remodeling.	[23]
	OPNc	1. OPNc was shown to exhibit a more significant enhancement of macrophage migration compared to OPNa in post-ischemic neovascularization.	[21]
		2. The activation of cellular calcium signals and subsequent nuclear translocation of nuclear factor of activated T-cells, cytoplasmic 2, induced by secretory OPNc, increased the survival of NSCLC cells treated with cisplatin	[24]
		3. OPNc spliced isoform has been shown to contribute to ovarian cancer progression	[25]
		4. In NSCLC, OPNc reduced cell proliferation, colony formation, and invasion.	[22]
		5. Elevated expression of OPNc activated PI3K and NF-κB pathways in endometriotic cells resulting in various cellular changes	[23]
		6. OPNc has been shown to increase functional collateral vessel formation in vivo	[21]
Post-translational modifications	Phosphorylation	Phosphorylation of OPN is a prerequisite for inducing IL-12 expression in macrophages, and its dephosphorylation nullifies this effect.	[26]
	Glycosylation	The folding structure, proteolytic cleavage, and functional characteristics of OPN are influenced by the presence of glycosylation. Removing several O-glycosylation sites from OPN affects cell adhesion activity and phosphorylation status.	[27,28]
	Transglutamination	Transglutaminase 2, a calcium-dependent enzyme, can utilize OPN as a substrate and catalyze the cross-linking of glutamine and lysine residues. This process can enable polymeric OPN to bind to the α9β1 receptor without relying on the SVVYGLR sequence.	[29,30]
Proteolytic cleavage	ntOPN	1. ntOPN has been shown to be associated with greater degrees of inflammation in carotid plaques in patients with hypertension	[31]
		2. ntOPN promotes abdominal aortic aneurysm by increasing the expression of pyroptosis-related inflammatory factors through the NF-κB pathway, inflammation, and extracellular matrix degradation.	[32]
		3. ntOPN controls activation of hepatic stellate cells and is essential for liver fibrogenesis.	[33]
	ctOPN	1. Studies have revealed that modifying the ctOPN can regulate its interaction with the widely expressed αVβ3-integrin.	[25]
		2. In vitro studies have shown that thrombin cleaved ctOPN can affect the migration and invasion of breast cancer cells.	[34]
		3. The ctOPN is reported to be involved in macrophage chemotaxis	[35]

OPN—Osteopontin; ntOPN—N-terminal OPN, ctOPN—C-terminal OPN, PI-3K—phosphotidylinositol-3 kinase, NF-κB—Nuclear factor kappa B.

**Table 2 biomedicines-11-01356-t002:** Role of osteopontin in various forms of chronic kidney disease.

Forms of Chronic Kidney Disease	Main Findings	References
Diabetic kidney disease (DKD)	1. High expression of OPN was reported in the tubular epithelium of the renal cortex and in glomeruli in rat and mouse models of diabetic nephropathy.	[72,73]
	2. OPN deletion prevented disease progression while OPN expression increased glomerular damage, possibly through the production of transforming growth factor-β, indicating that OPN might be a therapeutic target.	[65]
	3. OPN overexpression and macrophage recruitment were associated with considerable macrophage buildup in the renal interstitium in diabetic nephropathy, which may also be contributing to the tubulo-interstitial damage.	[72,74]
	4. Long-acting angiotensin-converting enzyme inhibitor, perindopril, significantly decreased the accumulation of macrophages and the expression of OPN, induced by diabetes, in the renal interstitium of diabetic rats.	[72]
	5. Blockade of the renin-angiotensin system by ramipril may confer renoprotection by decreasing OPN expression in non-insulin-dependent diabetic nephropathy.	[75]
Lupus Nephritis (LN)	1. A significant difference in the plasma concentration of OPN was seen in patients with systemic lupus erythematosus (SLE) and kidney impairment compared to healthy controls. The differences correlated with the levels of IL-18 and the SLE Disease Activity Index.	[76]
	2. Full-length OPN and ntOPN concentrations have been reported to be considerably greater in patients with LN. However, urine ntOPN was related to renal inflammation and thought to be a reliable prognostic indicator for LN.	[60,77]
	3. It was reported that intrarenal macrophage infiltration and higher OPN expression were positively associated in patients with LN.	[77]
Immunoglobulin A Nephropathy (IgAN)	1. Studies suggest that OPN is involved in the development of IgAN	[59]
	2. OPN had an accuracy of 87% in distinguishing IgAN from other glomerulopathies, and thus appeared to be a valuable biomarker.	[66]
	3. In this pathology both, OPN and CD44 receptor are highly expressed in cells in areas of tubulointerstitial injury.	[78]
	4. Another study conducted in children with IgAN showed that the high urinary level of OPN was associated with high OPN-to-creatinine ratio.	[79]
	5. Studies have shown that during the development of IgAN, increased OPN mRNA correlated with macrophage infiltration.	[79]
Autosomal dominant polycystic kidney disease (ADPKD)	1. OPN has been identified as a urinary biomarker for autosomal dominant polycystic kidney disease progression.	[80]
	2. The urinary OPN excretion levels were reported to be lower in rapid progressors than in slow progressors, suggesting that it may be a useful urinary biomarker for predicting ADPKD progression.	[80]
Minimal change disease (MCD)	1. A study in children with MCD revealed higher urinary OPN-to-creatinine ratio compared to the control group.	[79]
	2. A positive correlation of OPN mRNA expression in proximal tubules, and urinary OPN were observed in patients with MCD.	[59]
	3. Patients with MCD have higher urinary ntOPN, which was also associated with higher levels of albuminuria.	[59]
Membranous glomerulonephritis (MGN)	1. Recent studies demonstrated higher expression of OPN in the proximal tubules in patients with progressive and nonprogressive MGN.	[81,82]
	2. In this kidney disease, a strong correlation between the mRNA and OPN has been demonstrated. In a murine model, high expression of OPN in the kidney was associated with increased infiltration of macrophages and other immune cells, like CD4+ and CD8+ T lymphocytes.	[82]
	3. Overexpression of OPN in the kidney was also correlated with activation of NF-ĸB, increasing the expression of proinflammatory cytokines, which can contribute to glomerular damage.	[82]

OPN—osteopontin, CKD—chronic kidney disease; NF-κB—nuclear factor kappa B.

## Data Availability

Not applicable.

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
