# Peer review of "Osteopontin as a Biomarker in Chronic Kidney Disease"

_biomedicines, 2023, doi:10.3390/biomedicines11051356_

Round 1

Reviewer 1 Report

The reviewer thanks the authors for this incredibly well written and comprehensive review about a really mystifying subject! The question about OPN is always, what does it really do?

This reviewer has only one suggestion.  The authors review the expression patterns, spliced variants, post-translational modifications, etc.  A table or other format where the authors could list the varied forms of OPN, how the spliced variants and PTMs confer different properties and under what circumstances those might be important in health and disease would be great, if this is feasible.

Reviewer 2 Report

This is a nice review that provides up-to-dated information on the role of osteopontin as a biomarker in patients with CKD.

One small suggestion to the authors: please discuss evidence from studies in end-stage kidney disease (PMID: 35330148, PMID: 34468976, PMID: 31153055).

Reviewer 3 Report

This is a nice paper. However, I have some comments. The findings from this paper are excellent and worthy to review. This manuscript contained some questions described below. I think this paper is interesting, this review contributes to future's clinical medicine largely. I have some questions from a point of view of clinical medicine. In my opinion, this paper is of very high quality and is useful not only for basic research but also for clinical purposes. The role of osteopontin in the kidney is also well understood. I believe that it may have therapeutic applications in the future. On the other hand, I think we also need to consider the changes at the interstitial tubular level with regard to renal damage, especially fibrosis. Osteopontin seems to act mainly on the glomerulus, but what about its effect on the interstitial tubules? Please let me know if you have data on this. Or how about its association with tubular damage markers such as urinary beta2-microglobulin, NAG, NGAL, and L-FABP? Please add to the above and let us know.

Reviewer 4 Report

Osteopontin is a glycoprotein expressed in several tissues. Particularly in the kidney is expressed in the of loop Henle and in distal tubules. However, osteopontin is upregulated after renal damage in all tubular segments and the glomeruli, so it can be used as a renal damage biomarker. The authors describe the genetic structure of osteopontin and also describe and analyze the main findings of the changes of osteopontin in several diseases such as diabetic kidney disease, lupus nephritis, immunoglobulin A nephropathy, autosomal dominant polycystic kidney disease, and membranous glomerulonephritis as well as the therapeutic approaches to modulate osteopontin. The authors carried out an adequate bibliographical review and wrote the manuscript coherently and clearly. However, the following  points must be addressed

Minor points

  1. Line 51. Please indicate any references that demonstrate the role of osteopontin in the development and progression of diabetic nephropathy. I understood that is one of the works of the authors. However, there must be any finding related.
  2. Lines 76 and 93. The letter in Figure 1 must be written in capital letters.
  3. Figure legend 1. Lines 78-86. Some text parts are marked and should not be; please check. In addition, the abbreviations presented in the image as an aa, PI3K and mTOR, for example, should be defined at the final of this figure legend. 
  4. Line 117. There is an extra period before the word “motif”.
  5. Figure 2. The authors could indicate that it is plasma or urinary osteopontin. 
  6. Line 148. There is an extra period after the word “however”.
  7. Line 196. It should say Food and Drug Administration. 
  8. Line 303. The abbreviation of kilodalton is kDa, not kD; please correct it. 
  9. Table 1 is presented in the manuscript but not in the text; all figures and tables should be mentioned in the text to give continuity and meaning to the ideas presented.
  10. In the conclusion section, the authors could include a brief paragraph stating the methods for detecting osteopontin and nt-osteopontin in urine and plasma and the advantages over the other commonly used markers.
